# Variational Structured Semantic Inference for Diverse Image Captioning

**Fuhai Chen**[1], **Rongrong Ji**[12]*, **Jiayi Ji**[1], **Xiaoshuai Sun**[1], **Baochang Zhang**[3], **Xuri Ge**[1],
**Yongjian Wu**[4], **Feiyue Huang**[4], **Yan Wang**[5]
[1]Department of Artificial Intelligence, School of Informatics, Xiamen University,
[2]Peng Cheng Lab, [3]Beihang University, [4]Tencent Youtu Lab, [5]Pinterest
{cfh3c.xmu,jjyxmu,xurigexmu}@gmail.com, {rrji,xssun}@xmu.edu.cn, bczhang@buaa.edu.cn,
{littlekenwu,garyhuang}@tencent.com, yanw@pinterest.com

## Abstract

Despite the exciting progress in image captioning, generating diverse captions for a given image remains as an open problem. Existing methods typically apply generative models such as Variational Auto-Encoder to diversify the captions, which however neglect two key factors of diverse expression, *i.e.*, the lexical diversity and the syntactic diversity. To model these two inherent diversities in image captioning, we propose a Variational Structured Semantic Inferring model (termed VSSI-cap) executed in a novel structured encoder-inferer-decoder schema. VSSI-cap mainly innovates in a novel structure, *i.e.*, Variational Multi-modal Inferring tree (termed VarMI-tree). In particular, conditioned on the visual-textual features from the encoder, the VarMI-tree models the lexical and syntactic diversities by inferring their latent variables (with variations) in an approximate posterior inference guided by a visual semantic prior. Then, a reconstruction loss and the posterior-prior KL-divergence are jointly estimated to optimize the VSSI-cap model. Finally, diverse captions are generated upon the visual features and the latent variables from this structured encoder-inferer-decoder model. Experiments on the benchmark dataset show that the proposed VSSI-cap achieves significant improvements over the state-of-the-arts.

## 1  Introduction

Image captioning has recently attracted extensive research attention with broad application prospects. Most state-of-the-art image captioning models adopt an encoder-decoder architecture [1, 2, 3], which encodes the image into a feature representation via Convolutional Neural Network (CNN) and then decodes the feature into a caption via Recurrent Neural Networks with Long-Short Term Memory units (LSTM). Despite the exciting progress, one common defect is that the generated captions are semantically synonymous and syntactically similar, which goes against the inherent diversity delivered by the image, *i.e.*, "*A picture is worth a thousand words*". Nevertheless, generating diverse captions from a given image remains as an open problem. As shown in Fig. 1 (Left-Top), it is quite intuitive to derive heterogeneous understanding from human being, while the traditional models typically tend to generate homogeneous sentences due to the limited variation in the maximum likelihood objective [4].

Several recent works have been proposed to investigate diverse image captioning [5, 6, 7, 8, 9], which typically employed a Generative Adversarial Network (GAN) or Variational Auto-Encoder (VAE) as the generative model. For example, [5] designed an adversarial model trained with an approximate sampler to implicitly match the generated distribution to the human caption. For another instance, [8] proposed a conditional VAE based captioning model guided by an object-wise prior, as roughly

---

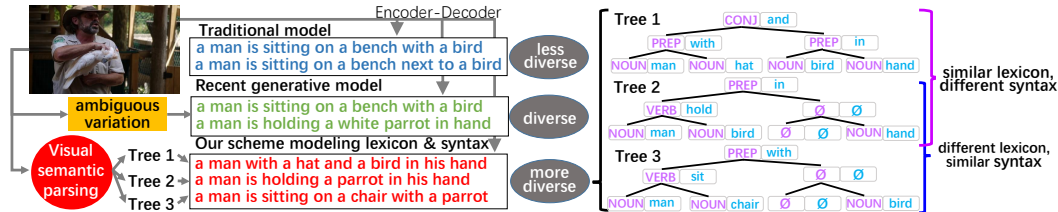

Figure 1: Illustration of diverse image captioning. Left: Captions generated by traditional image captioning model (Left-Top), state-of-the-art generative model (Left-Middle), and our scheme that explicitly models lexical and syntactic diversities (Left-Bottom) for diverse image captioning. Right: Captions with higher diversity are generated when the lexical (light blue) and syntactic (purple) diversities are considered.

shown in Fig. 1 (Left-Middle). However, all these methods treated diverse image captioning as a blackbox without explicitly modeling the key factors to diversify the expression, *i.e.*, the lexical and syntactic diversities, as revealed in the natural language research [10, 11, 12], which in principle involves identifying content entities and then expressing their relationships. Fig. 1 (Left-Bottom and Right) shows an example of the lexical and syntactic diversities, both of which should be taken into account for generating diverse image captions.

In this paper, we aim at explicitly modeling the lexical and syntactic diversities from the visual content towards diversified image caption generation. To this end, we tackle two fundamental challenges, *i.e.*, *diversity modeling* and *diversity embedding*. For diversity modeling, we infer the lexical and syntactic variables from the visual content by leveraging the visual parsing tree (VP-tree) [13, 14, 15, 16], which predicts the probability distributions of the lexical and syntactic categories to weight the latent variables in variational inferences. For diversity embedding, we advance the commonly-used encoder-decoder scheme into a new *structured encoder-inferer-decoder* scheme, where the aforementioned variational inference is treated as an inferer and its outputs, *i.e.*, the lexical and syntactic latent variables (with variations), are sampled together with visual features to feed a LSTM-based caption generator.

In particular, we propose a novel **V**ariational **S**tructured **S**emantic **I**nferring model for diverse image captioning, termed VSSI-cap as illustrated in Fig. 2, which is deployed over VAE[2] to model and embed the lexical and syntactic diversities. In general, towards diversity modeling, such diversities are inferred in the designed variational multi-modal inferring tree (termed, VarMI-tree). Towards diversity embedding, such diversities are integrated into diverse image captioning in a new structured encoder-inferer-decoder scheme. In particular, the proposed model contains three components: 1) **encoder**: Given an image and its corresponding caption, the visual and textual features are extracted by CNN and a word embedding model, respectively. 2) **inferer**: Inspired by the recent work in visual semantic parsing [13], a VarMI-tree is proposed to infer the latent variables with variations for the lexical and syntactic diversities. 3) **decoder**: The visual feature, the inferred lexical and syntactic variables (from posterior/prior inference), are decoded to output the caption by using LSTM.

The contributions of this paper are as follows: 1) We are the first to explicitly model diverse image captioning based on the lexical and syntactic diversities. We address two key issues in diverse captioning, *i.e.* diversity modeling and diversity embedding. 2) For diversity modeling, we propose a novel variational multi-modal inferring tree (VarMI-tree) to model the lexical and syntactic diversities. 3) For diversity embedding, we propose a structured encoder-inferer-decoder scheme which explicitly integrates the lexical and syntactic diversities in caption generation. 4) The proposed VSSI-Cap beats the state-of-the-arts [5, 8] on the MSCOCO benchmark dataset in terms of both accuracy metrics and diversity metrics.

## 2 Preliminary

**Image Captioning.** We adopt an encoder-decoder architecture as the basic image captioning model, where CNN is employed to encode an image $I$ into a deep visual feature $\mathbf{v}$ and LSTM is used to decode this visual feature into a caption $S$. Many state-of-the-art methods [2, 3] adopt a maximum likelihood principle to train the models by using the image-caption pair set

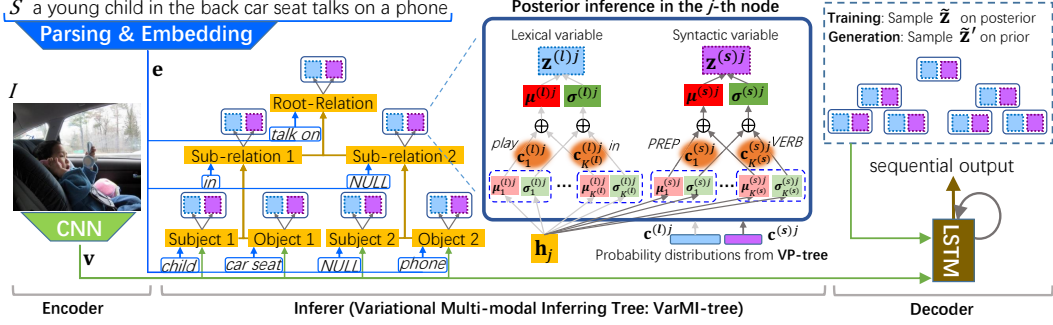

Figure 2: Overview of the proposed VSSI-cap model for diverse image captioning, which consists of *encoder*, *inferer*, and *decoder*. Given an image and its corresponding caption during training, visual feature $\mathbf{v}$ and textual feature $\mathbf{e}$ are extracted from CNN and word embedding model respectively in **encoder** (brief in Sec. 3.2). In **inferer** (Sec. 3.2), to represent the lexical/syntactic diversity, a VarMI-tree is designed to infer the latent lexical/syntactic variable $\mathbf{z}^{(\ell)}/\mathbf{z}^{(s)}$ upon an additive Gaussian distribution in each node, where the means $\boldsymbol{\mu}_{1:K}$ and the square deviations (sds) $\boldsymbol{\sigma}_{1:K}$ over different lexical/syntactic components are parameterized upon the node feature $\mathbf{h}$, and subsequently weighted by the corresponding probability distributions $\mathbf{c}_{1:K}$ from the VP-tree (Sec. 3.1) for the additive Gaussian parameters $\boldsymbol{\mu}$ and $\boldsymbol{\sigma}$. In **decoder** (Sec. 3.3), $\tilde{\mathbf{z}}$ is sampled from the posterior inference and is used for training while $\tilde{\mathbf{z}}'$ is sampled from the prior inference (similar to the posterior inference but with $\boldsymbol{\mu}_{1:K}$ and $\boldsymbol{\sigma}_{1:K}$ initialized randomly, detailed in Sec. 3.2) and is used to generate captions. Finally, $\mathbf{v}$ and $\tilde{\mathbf{z}}/\tilde{\mathbf{z}}'$ are fed into LSTM for the sequential caption outputs.

$\mathcal{D}_p = \{I^{(i)}, S^{(i)} = \{S_t^{(i)}\}_{t=0}^{T^{(i)}}\}_{i=0}^{N_p}$, where $N_p$ and $T$ denote the pair number and the caption length, respectively. The corresponding objective function can be formulated as follows:

$$\log P(S|I) = \frac{1}{N_p} \sum_{i=0}^{N_p} \sum_{t=0}^{T_i} \log p(S_t^{(i)}|\mathbf{v}^{(i)}, S_{0:t-1}^{(i)}). \tag{1}$$

However, the above schemes are unsuitable for generating multiple diverse caption candidates due to the certainty of encoding in Eq. 1. Therefore, generative models, such as GAN and VAE, are typically exploited to handle diverse image captioning [5, 6, 7, 8]. Other related topics include: personalized expression [17, 18], stylistic description [19, 20], online context-aware heuristic search [21, 22], and word-specific discriminative captioning [15] *etc*.

**Variational Auto-Encoder (VAE).** We briefly present the variational auto-encoder (VAE) [23, 24] and its conditional variant [25, 26], which serves as the fundamental framework of the proposed structured encoder-inferer-decoder scheme. Given an observed variable $x$, VAEs aim at modeling the data likelihood $p(x)$ based on the assumption that $x$ is generated from a latent variable $z$, *i.e.*, the *decoder* $p(x|z)$, which is typically estimated via deep nets. Since the posterior inference $p(z|x)$ is not computably tractable, it is approximated with a posterior inference $q(z|x)$ that is typically a diagonal Gaussian $\mathcal{N}(\mu, diag(\sigma^2))$, where the mean $\mu$ and the square deviation $\sigma$ can be parameterized in deep nets and serve as the *encoder*[3]. Thus, the *encoder/inferer* and *decoder* can be optimized by maximizing the following lower bound:

$$\mathcal{L}_{\text{VAE}}(\theta, \phi; x, c) = \mathbb{E}_{q_\phi(z|x,c)}\big[\log p_\theta(x|z, c)\big] - D_{\text{KL}}\big(q_\phi(z|x,c)\|p_\theta(z|c)\big) \leq \log p_\theta(x), \tag{2}$$

where $\mathbb{E}$ and $D_{\text{KL}}$ are the approximate expectation and Kullback-Leibler (KL) divergence, respectively. $c$ denotes the condition, which exists in the case of conditional VAE (CVAE). $\phi$ and $\theta$ denote the parameters of the *inferer* and the *decoder* (*e.g.*, LSTM), respectively. For diverse image captioning, it's a straightforward thinking to represent the visual feature and the caption with $c$ and $x$ respectively in a VAE model. However, the latent variable $z$ in such VAE model has a very general prior (standard Gaussians), which does not consider any domain-specific knowledge. We argue that it may waste model capacity, and one should consider the unique problem structures of image captioning instead of using the VAE as is.

Table 1: Main notations and their definitions.

| Notation | Definition |
|---|---|
| $I$ | an image |
| $S$ | a caption |
| $\mathbf{v}$ | the visual feature |
| $\mathbf{e}_j$ | the $j$-node word embedding feature |
| $\mathbf{h}_j$ | the feature of the $j$-th node in VarMI-tree |
| $\mathbf{z}^{(\ell)j}/\mathbf{z}^{(s)j}$ | the $j$-node lexical/syntactic latent variable |
| $\mathbf{c}^{(\ell)j}/\mathbf{c}^{(s)j}$ | the $j$-node lexical(word's)/syntactic(POS's) probability distribution in VP-tree |
| $\boldsymbol{\mu}^{(\ell)j}/\boldsymbol{\mu}^{(s)j}$ | the additive mean of the $j$-node lexical/syntactic posterior Gaussian distribution |
| $\boldsymbol{\sigma}^{(\ell)j}/\boldsymbol{\sigma}^{(s)j}$ | the additive squ. dev. of the $j$-node lexical/syntactic posterior Gaussian distribution |
| $\boldsymbol{\mu}_k^{(\ell)j}/\boldsymbol{\mu}_k^{(s)j}$ | the $k$-component mean of the $j$-node lexical/syntactic posterior Gaussian distribution |
| $\boldsymbol{\sigma}_k^{(\ell)j}/\boldsymbol{\sigma}_k^{(s)j}$ | the $k$-component squ. dev. of the $j$-node lexical/syntactic posterior Gaussian distribution |
| $\theta$ | the parameter set of the decoder |
| $\phi^{(\ell)}/\phi^{(s)}$ | the lexical/syntactic parameter set of the inferer |
| $\psi$ | the parameter set of VarMI-tree trunk |
| $'$ | the mark for the prior |

## 3 The Proposed VSSI-Cap Model

The framework of the proposed VSSI-Cap model is illustrated in Fig. 2. Following Eq. 2, the model is in principle optimized by maximizing the lower bound on the log-likelihood of $p_\theta(S)$ as below:

$$
\mathcal{L}(\theta, \phi^{(\ell)}, \phi^{(s)}, \psi; S, \mathbf{v}, \mathbf{c}^{(\ell)}, \mathbf{c}^{(s)}) = \mathbb{E}_{\mathbf{z}^{(\ell)} \sim q_{\phi^{(\ell)}, \psi}, \mathbf{z}^{(s)} \sim q_{\phi^{(s)}, \psi}} \left[ \log p_\theta(S | \mathbf{z}^{(\ell)}, \mathbf{z}^{(s)}, \mathbf{v}, \mathbf{c}^{(\ell)}, \mathbf{c}^{(s)}) \right]
$$

$$
- D_{\mathrm{KL}}\big(q_{\phi^{(\ell)}, \psi}(\mathbf{z}^{(\ell)} | S, \mathbf{v}, \mathbf{c}^{(\ell)}) \| p(\mathbf{z}^{(\ell)} | \mathbf{c}^{(\ell)})\big) - D_{\mathrm{KL}}\big(q_{\phi^{(s)}, \psi}(\mathbf{z}^{(s)} | S, \mathbf{v}, \mathbf{c}^{(s)}) \| p(\mathbf{z}^{(s)} | \mathbf{c}^{(s)})\big), \quad (3)
$$

which consists of two components, *i.e.*, the approximate expectation $\mathbb{E}$ and the KL divergence $D_{\mathrm{KL}}$. The former is maximized to reduce the reconstruction loss of the caption generation in decoder as Eq. 1, while the later measures the difference between the distributions of the posterior $q_{\phi,\psi}(\mathbf{z}|S, \mathbf{v}, \mathbf{c})$ and the prior $p(\mathbf{z}|\mathbf{c})$ for the prior guidance (detailed in Sec. 3.3). **Firstly**, we define the variables and parameters as following: "$(\ell)$" and "$(s)$" are the marks for the variables and parameters of the lexicon and the syntax respectively. $\mathbf{v}$, $\mathbf{z}$, and $\mathbf{c}$ denote the visual feature of the image $I$, the lexical/syntactic latent variable, and the lexical(word's)/syntactic(POS's) probability distribution from VP-tree (see Fig. 3), respectively. $S$ denotes the caption, which is parsed and embedded into the textual feature $\mathbf{e}$.[4] $\theta$ is the parameter set in the *decoder*, while $\phi$ and $\psi$ are the parameter sets of the lexical/syntactic posterior inference and the VarMI-tree trunk, respectively, in the *inferer*. **Secondly**, we introduce the posterior/prior (Sec. 3.2) based on the above definitions: we adopt an additive Gaussian distribution for the posterior/prior to infer the latent variables. As shown in the middle of Fig. 2, the additive parameters, *i.e.*, the mean $\boldsymbol{\mu}/\boldsymbol{\mu}'$ and the square deviation (sd) $\boldsymbol{\sigma}/\boldsymbol{\sigma}'$, are derived from multiple component parameters (means and sds of multiple Gaussian distributions, corresponding to different word's and POS's components and weighted by the probability distributions). **Thirdly**, we describe the posterior/prior inference (Sec. 3.2): In the posterior inference, the additive and component parameters are both parameterized by a linear function, while the component parameters are initialized randomly in the prior inference as shown in the middle of Fig. 2. Here we omit the prior inference due to the similarity to the posterior inference. The corresponding lexical-syntactic latent variable $\tilde{\mathbf{z}}/\tilde{\mathbf{z}}'$ is sampled from the posterior/prior inference by reparameterizing $\mathbf{z}/\mathbf{z}'$. **Finally**, during training/generation (detailed in Sec. 3.3), the visual feature $\mathbf{v}$ and the latent variable $\tilde{\mathbf{z}}/\tilde{\mathbf{z}}'$ are fed into LSTM to generate sequential caption outputs.

In the following, we briefly introduce the lexicon-syntax based VP-tree in Sec. 3.1. We then give the details about the proposed VarMI-tree in Sec. 3.2. Finally, in Sec. 3.3, we introduce the proposed structured encoder-inferer-decoder schema. For clarity, the main notations and their definitions throughout the paper are shown in Tab. 1.

## 3.1 Visual Parsing Tree

Visual parsing tree (VP-tree) is firstly proposed in [13], which serves as a robust parser to discover visual entities and their relations from a given image. To parse them in the lexicon and the syntax, we modify VP-tree as Fig. 3 (a), where the probability distributions of $K^{(\ell)}$ words and $K^{(s)}$ POSs, *i.e.*, $\mathbf{c}^{(\ell)} \in R^{K^{(\ell)}}$ and $\mathbf{c}^{(s)} \in R^{K^{(s)}}$, are estimated in each node for weighting in the subsequent VarMI-tree. There are $M$ (typically, $M = 7$) tree nodes in these two binary trees. To distinguish these two trees, we define the variables and parameters of VP-tree with " _ ". As shown in Fig. 3 (a), VP-tree consists of three operations, *Semantic mapping*, *Node combining*, and *Classifying*, where the first two adopt normal linear

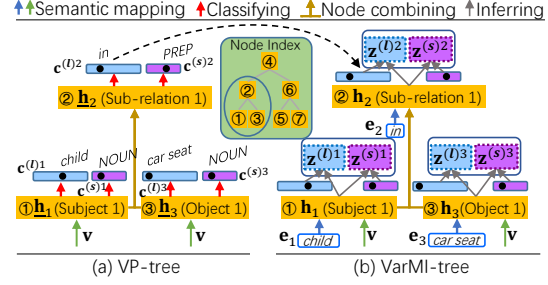

Figure 3: The examplar subtrees of VP-tree and VarMI-tree. The differences lie in: 1) Single *vs.* multi-modal *Semantic mapping*. 2) Whether inferring the lexical and syntactic latent variables $\mathbf{z}^{(\ell)}$ and $\mathbf{z}^{(s)}$ or not. 3) Probability distributions $\mathbf{c}^{(\ell)}$ and $\mathbf{c}^{(s)}$ of the optimized VP-tree are utilized for weighting in the inference of VarMI-tree.

mapping and concatenating operations upon visual feature $\mathbf{v}$ to obtain the node feature $\underline{h}$. In *Classifying*, the feature $\underline{h}$ of each node is mapped into the word's and POS's category spaces, respectively according to their vocabularies. For the $j$-th node, we obtain its word and POS probability distributions, *i.e.*, $\mathbf{c}^{(\ell)j}$ and $\mathbf{c}^{(s)j}$, as follows:

$$
\begin{aligned}
\mathbf{c}_r^{(\ell)j} = f^{(cl)}(\underline{\mathbf{W}}_r^{(\ell)}\underline{h}_j + \underline{b}_r^{(\ell)}), \quad \mathbf{c}_r^{(s)j} = f^{(cl)}(\underline{\mathbf{W}}_r^{(s)}\underline{h}_j + \underline{b}_r^{(s)}), \\
s.t.(j:r) \in \{(1:E),(3:E),(5:E),(7:E),(2:R),(6:R),(4:R)\},
\end{aligned}
\tag{4}
$$

where $f^{(cl)}$ is a Softmax function with parameters $\underline{\mathbf{W}}_r^{(cl)}$ and $\underline{b}_r^{(cl)}$ for the entity ($r =$"$E$") or the relation ($r =$"$R$") classifications. We unify $\mathbf{c}_r^j$ ($r =$"$E$","$R$") into $\mathbf{c}^j$ for simplification. During training, $\mathbf{c}^j$ is used to compute the cross entropy loss with the lexical/syntactic category labels[4]. The parameter set is finally optimized for automatic tree construction given an image feature, where each node provides the optimal word's and POS's probability distributions $\mathbf{c}^{(\ell)}$ and $\mathbf{c}^{(s)}$. Note that one can replace VP-Tree with other alternative visual structured representations for the lexicon and syntax. However, in order to directly demonstrate the effectiveness of the core idea, we intentionally chose the straightforward assistance of VP-Tree.

## 3.2 Variational Multi-modal Inferring Tree

The major challenge of VSSI-cap is to model the posterior inference of both the lexical and syntactic latent variables, *i.e.*, $q_{\phi^{(\ell)},\psi}(\mathbf{z}^{(\ell)}|S,\mathbf{v},\mathbf{c}^{(\ell)})$ and $q_{\phi^{(s)},\psi}(\mathbf{z}^{(s)}|S,\mathbf{v},\mathbf{c}^{(s)})$, in the tree structure. To this end, we design a variational multi-modal inferring tree (VarMI-tree) to further innovate the VP-tree as illustrated in Fig. 3 (b) and Fig. 2 (Middle). VarMI-tree consists of three operations, *i.e.*, *Semantic mapping*, *Node combining*, and *Inferring*. We itemize them as follows:

**Semantic Mapping.** In the encoder, the visual feature $\mathbf{v}$ is extracted from the last fully-connected layer of CNN [27] while the $j$-th word's feature $\mathbf{e}_j$ ($j \in \{1, \ldots, M\}$ corresponds to the $j$-th tree node) of the caption $S$ is extracted by textual parsing and word embedding as aforementioned. In the inferer, these features are mapped into different semantic spaces, *i.e.*, subjects, objects, and relations in VarMI-tree as shown in Fig. 3 and Fig. 2, which can be formulated as:

$$
\begin{aligned}
\mathbf{h}_j = f^{(sm)}\big(\mathbf{W}_r^{(sm)}[\mathbf{v};\mathbf{e}_j] + \mathbf{b}_r^{(sm)}\big), \\
s.t.(j:r) \in \{(1:Subj1),(3:Obj1),(5:Subj2),(7:Obj2)\},
\end{aligned}
\tag{5}
$$

where $r$ represents one of four semantic entity items, *i.e.*, subject 1, object 1, subject 2, and object 2 as set up in VP-tree. $[\cdot;\cdot]$ is the concatenation operation. $f^{(sm)}$ denotes a non-linear function with the parameters $W_r^{(sm)}$ and $b_r^{(sm)}$ for *Semantic mapping* in the $j$-th node ($j = 1,3,5,7$). For the non-leaf nodes ($j = 2,4,6$), similar operation is conducted as above, where, however, $\mathbf{v}$ is replaced with the combination features (computed in next part) as shown in Fig. 3.

**Node Combining.** The *Node combining* operation of VarMI-tree is the same as the one of VP-tree. Correspondingly, we denote the parameters with $\mathbf{W}^{(nc)}$ and $\mathbf{b}^{(nc)}$.

**Inferring.** For clarity, we define the function $H_j$ as a unified operation of the above *Semantic mapping* and *Node combining* for the $j$-th node feature, *i.e*, $\mathbf{h}_j = H_j(\mathbf{v}, \mathbf{e}; \psi)$. In the $j$-th node, the lexical and syntactic posterior inferences can be approximated upon an additive Gaussian distribution. For clarity, we only formulate it on the lexicon below:

$$q_{\phi^{(\ell)},\psi}(\mathbf{z}^{(\ell)j}|S,\mathbf{v},\mathbf{c}^{(\ell)j}) = \mathcal{N}\big(\mathbf{z}^{(\ell)j}\big|\sum_{k=1}^{K^{(\ell)}} c_k^{(\ell)j}\boldsymbol{\mu}_k^{(\ell)j}(H_j), \boldsymbol{\Sigma}^{(\ell)j2}\mathbb{I}\big), \tag{6}$$

where $\boldsymbol{\Sigma}^{(\ell)j2}\mathbb{I}$ is the spherical covariance matrix with $\boldsymbol{\Sigma}^{(\ell)j2} = \sum_{k=1}^{K^{(\ell)}} c_k^{(\ell)j}\boldsymbol{\sigma}_k^{(\ell)j}(H_j)^2$. $K^{(\ell)}$ denotes the length of the word's vocabulary. The component Gaussian parameters can be obtained:

$$\boldsymbol{\mu}_k^{(\ell)j}(H_j) = \mathbf{W}_{\mu_k}^{(\ell)j}H_j + \mathbf{b}_{\mu_k}^{(\ell)j}, \quad \log\boldsymbol{\sigma}_k^{(\ell)j}(H_j)^2 = \mathbf{W}_{\sigma_k}^{(\ell)j}H_j + \mathbf{b}_{\sigma_k}^{(\ell)j}, \tag{7}$$

To enable the differentiability in the end-to-end manner, we reparameterize $\mathbf{z}^{(\ell)j}$ into $\tilde{\mathbf{z}}^{(\ell)j}$ via the reparameterization trick [23] as:

$$\tilde{\mathbf{z}}^{(\ell)j} = \boldsymbol{\mu}^{(\ell)j} + \boldsymbol{\sigma}^{(\ell)} \odot \boldsymbol{\varepsilon}^{(\ell)}, \tag{8}$$

where $\boldsymbol{\varepsilon}^{(\ell)}$ obeys a standard Gaussian distribution to introduce noise for the lexical diversity. $\odot$ is an element-wise product. Similar to the posterior, the prior $p(\mathbf{z}^{(\ell)j}|\mathbf{c}^{(\ell)j})$ can be formulated as:

$$p(\mathbf{z}^{(\ell)j}|\mathbf{c}^{(\ell)j}) = \mathcal{N}\big(\mathbf{z}^{(\ell)j}\big|\sum_{k=1}^{K^{(\ell)}} c_k^{(\ell)j}\boldsymbol{\mu}_k'^{(\ell)j}, (\sum_{k=1}^{K^{(\ell)}} c_k^{(\ell)j}\boldsymbol{\sigma}_k'^{(\ell)j2})\mathbb{I}\big), \tag{9}$$

where $\boldsymbol{\mu}_k'^{(\ell)}$ and $\boldsymbol{\sigma}_k'^{(\ell)}$ are randomly initialized. $\mathbf{z}'^{(\ell)j}$ is reparameterized into $\tilde{\mathbf{z}}'^{(\ell)j}$ as Eq. 8.

### 3.3 Structured Encoder-inferer-decoder

The structured encoder-inferer-decoder schema aims at integrating the lexical/syntactic latent variables in a tree structure to diversify the generated captions. Following Eq. 3, we give the final objective function as follows:

$$\mathcal{L}_{\text{VSSI-Cap}}(\theta, \phi^{(\ell)}, \phi^{(s)}, \psi; S, \mathbf{v}, \mathbf{c}^{(\ell)}, \mathbf{c}^{(s)}) = \mathbb{E}_d(\theta; S, \mathbf{v}, \mathbf{c}^{(\ell)}, \mathbf{c}^{(s)})$$
$$-\sum_{j=1}^{M}\Big[D_{\text{KL}}\big(q_{\phi^{(\ell)},\psi}(\mathbf{z}^{(\ell)j}|S,\mathbf{v},\mathbf{c}^{(\ell)j})\|p(\mathbf{z}^{(\ell)j}|\mathbf{c}^{(\ell)j})\big) + D_{\text{KL}}\big(q_{\phi^{(s)},\psi}(\mathbf{z}^{(s)j}|S,\mathbf{v},\mathbf{c}^{(s)j})\|p(\mathbf{z}^{(s)j}|\mathbf{c}^{(s)j})\big)\Big], \tag{10}$$

where most of the above notations are defined in Eq. 3. $D_{\text{KL}}$ can be approximated following [28] (see algorithm flow in supplementary material). $\mathbb{E}_d$ is the approximate expectation on the log-likelihood of $p_\theta(S|I)$ in *decoder*. For the reconstruction loss, we use Monte Carlo method to approximate the expectation $\mathbb{E}_d$ in Eq. 10 after sampling $\tilde{\mathbf{z}}^{(\ell)j}$ and $\tilde{\mathbf{z}}^{(s)j}$, which is formulated as:

$$\mathbb{E}_d = \frac{1}{N}\sum_{i=1}^{N}\sum_{t=0}^{T}\log p(S_t|S_{0:t-1}, \{\mathbf{z}^{(\ell)j(i)}\}_{j=1}^{M}, \{\mathbf{z}^{(s)j(i)}\}_{j=1}^{M}, \mathbf{v}, \mathbf{c}^{(\ell)}, \mathbf{c}^{(s)}), \tag{11}$$
$$s.t.\ \forall i,j\ \mathbf{z}^{(\ell)j(i)} \sim q_{\phi^{(\ell)},\psi}(\mathbf{z}^{(\ell)j}|S,\mathbf{v},\mathbf{c}^{(\ell)j}), \mathbf{z}^{(s)j(i)} \sim q_{\phi^{(s)},\psi}(\mathbf{z}^{(s)j}|S,\mathbf{v},\mathbf{c}^{(s)j}),$$

where $N$ and $T$ denote the sample number of $\mathbf{z}^{(i)}$ (sampled by Eq. 8) and the length of the caption, respectively. Since the objective function in Eq. 10 is differentiable, we optimize the model parameter set $\theta$, $\phi^{(\ell)}$, $\phi^{(s)}$, and $\psi$ jointly using stochastic gradient ascent method. To generate captions, we use the above optimal parameters and choose the $t$-th word $\tilde{S}_t$ over the dictionary according to $\tilde{S}_t = \arg\max_{S_t} p(S_t|S_{0:t-1}, \mathbf{z}'^{(\ell)}, \mathbf{z}'^{(s)}, \mathbf{v})$, where $\mathbf{v}$ and $\mathbf{z}'$ are concatenated to feed the decoder.

## 4 Experiments

**Dataset and Metrics.** We conduct all the experiments on the MSCOCO dataset[5] [30], which is widely used for image captioning [1, 3] and diverse image captioning [5, 8]. There are over 93K images in MSCOCO, which has been split into training, testing and validating sets[6]. Each image has at least five manual captions. The quality of captioning results lies in both accuracy (a basic evaluation of captioning quality and has been used together with the subsequent diversity metrics in [8, 5, 6]) and diversity. For accuracy, we use the MSCOCO caption evaluation tool[7] by choosing

Table 2: Performance comparisons on accuracy of diverse image captioning. All values are in %. The first and the second places are marked with the bold font and "__" respectively.

| Metric | Bleu-1 | Bleu-2 | Bleu-3 | Bleu-4 | Meteor | Rouge-L | CIDEr | Spice |
|---|---|---|---|---|---|---|---|---|
| ErDr-cap [29] | 69.9 | 51.8 | 36.6 | 25.6 | 23.1 | 50.3 | 84.3 | 16.4 |
| Up-Down [3] | **79.8** | - | - | **36.3** | **27.7** | **56.9** | **120.1** | **21.4** |
| G-GAN [6] | - | - | 30.5 | 20.7 | 22.4 | 47.5 | 79.5 | _18.2_ |
| Adv [5] | - | - | - | - | 23.9 | - | - | 16.7 |
| CAL [9] | 66.5 | 48.4 | 33.2 | 21.8 | 22.6 | 47.8 | 75.3 | 16.4 |
| GMM-CVAE [8] | 70.0 | 52.0 | 37.1 | 26.0 | 23.2 | 50.6 | 85.4 | 16.3 |
| AG-CVAE [8] | 70.2 | 52.2 | 37.1 | 26.0 | 23.4 | 50.6 | 85.7 | 16.5 |
| VSSI-cap-L | 69.9 | 51.9 | 37.3 | 26.1 | 23.5 | 50.7 | 87.3 | 16.8 |
| VSSI-cap-S | 70.4 | 52.7 | 37.9 | 27.1 | 23.8 | 51.1 | 88.8 | 17.0 |
| VSSI-cap | _70.4_ | _52.7_ | _38.1_ | _27.3_ | _23.9_ | _51.3_ | _89.4_ | 17.1 |

Table 3: Performance comparisons on diversity. "⇓" and "⇑" denote that lower and higher are better, respectively. "n" denotes the number of generated captions (default 5). All values are in %. The first and the second places are marked with the bold font and "__" respectively.

| Metric | Num. | mB.⇓ | div1⇑ | div2⇑ | Uni.⇑ | Nov.⇑ |
|---|---|---|---|---|---|---|
| human | n=5 | **51.0** | **34.0** | **48.0** | **99.8** | - |
| ErDr-cap [29] | n=5 | 78.0 | 28.0 | 38.0 | - | 34.18 |
| Up-Down [3] | n=5 | 80.9 | 27.1 | 35.8 | - | 63.60 |
| G-GAN [6] | n=5 | - | - | - | - | 81.52 |
| Adv [5] | n=5 | 70.0 | _34.0_ | 44.0 | - | 73.92 |
| CAL [9] | n=5 | - | 32.5 | 40.7 | - | - |
| AG-CVAE [8] | n=5 | 70.2 | 33.1 | 42.9 | 66.9 | 79.67 |
| AG-CVAE [8] | n=10 | 77.3 | 22.7 | 31.3 | 70.8 | 79.68 |
| VSSI-cap-L | n=5 | 68.7 | 34.3 | 45.6 | 80.2 | 79.30 |
| VSSI-cap-S | n=5 | 63.0 | 33.8 | 46.3 | 82.4 | 80.26 |
| VSSI-cap | n=5 | _62.4_ | 33.9 | _47.2_ | _83.0_ | _85.20_ |
| VSSI-cap | n=10 | 74.2 | 22.3 | 33.2 | 80.7 | 80.34 |

the best-performing one from the top-5 outputs, including Bleu, Meteor, Rouge-L, CIDEr [30] and Spice [31]. For diversity, we use the benchmark metrics in [5, 8]: 1) *Div1*, the ratio of unique unigrams to words in the generated captions. Higher div1 means more diverse. 2) *Div2*, the ratio of unique bigrams to words in the generated captions. Higher div2 means more diverse. 3) *mBleu* (mB.), the mean of Bleu scores, which are computed between each caption in the generated captions against the rest. Lower mB. means more diverse. 4) *Unique Sentence* (Uni.), the average percentage of unique captions in candidates generated for each image. 5) *Novel Sentence* (Nov.), the percentage of the generated captions that do not appear in the training set. For uniformity, each output caption corresponds to a sample of $\mathbf{z}$.

**Preprocessing, Parameter Settings, and Implementation Details.** In the proposed VarMI-tree, we set the feature dimension of each node as 512. The dimensions of each mean, each sd, and each latent variable are set as 150. We parse the captions by using the Stanford Parser [32] as well as pruning the textual parsing results by using the pos-tag tool and the lemmatizer tool in NTLK [33], where the dynamic textual parsing trees are converted to a fixed-structured, three-layer, complete binary tree as designed in [13]. Only the words (including entities and relations) and the POSs (*i.e.*, *NOUN*, *VERB*, *PREP*, and *CONJ*) with high frequency are left to form the vocabularies. Nouns are regarded as entities and used as leaf nodes in the textual parsing tree, while others (verbs, coverbs, prepositions, and conjunctions) are taken as relations for non-leaf nodes. The sizes of the entity's, relation's and POS's vocabularies, are 840, 248, and 4, respectively.[8] We extract the visual features from VGG-16 network [25]. In LSTM, we use the same vector dimensions of the hidden states as [29], which is set as 512. We set the word vector dimension as 256 during word embedding. We implement our model training based on the public code[9] with the standard data split and the separate $\mathbf{z}$ samples. KL annealing method [34] is adopted to reduce the KL vanishing (see the supplementary material for the training details). All networks are trained with SGD with a learning rate 0.005 for the first 5 epochs, and is reduced by half every 5 epochs. On average, all models converge within 50 epochs. The overall process takes 37 hours on a NVIDIA GeForce GTX 1080 Ti GPU with 11GB memory.

**Baselines and Competing Methods.** We compare the proposed VSSI-cap with four baselines: 1) *ErDr-cap*: a caption generator trained based on encoder-decoder (beam search) [29] that represents the mainstream of general image captioning. 2) *AG-CVAE* [8]: a recent generative model considering the variation over detected objects for diverse image captioning. 3) *VSSI-cap-L*: an alternative version of VSSI-cap, which omits the syntax. 4) *VSSI-cap-S*: an alternative version of VSSI-cap, which omits the lexicon. We also compare VSSI-cap with the state-of-the-art method *Adv* [5] and *AG-CVAE* [8] (evaluated on the aforementioned universal split). Besides, we compare VSSI-cap with 1) other recent diverse image captioning methods, including *G-GAN* [6], *GMM-CVAE* [8], and *CAL* [9], 2) the state-of-the-art image captioning method, *i.e.*, *Up-Down* (beam search) [3], and 3) *Human*: a sentence randomly sampled from ground-truth/manually-labeled annotations of each image is used as the output of this method. Note that comparing to pure image captioning methods (only aiming at accuracy) seems far-fetched due to the mutual interference between accuracy and diversity (a more diverse caption tends to be more inconsistent with the ground truth caption) [9, 6, 5], where, therefore, the pure image captioning methods are taken as extraessential references.

**Evaluation on Accuracy.** Tab. 2 presents the accuracy comparisons of our VSSI-cap to the baselines and state-of-the-arts. Compared to others (except the state-of-the-art image captioning method), VSSI-cap achieves the best performance under most metrics. Specially, VSSI-cap outperforms AG-CVAE under all metrics, *e.g.*, 89.4% vs. 85.7% on CIDEr, which reflects the superiority of visual semantic representation in the proposed VarMI-tree. Additionally, the propsoed structured encoder-inferer-decoder schema also contributes to the improvement of accuracy according to the comparison with ErDr-cap. Particularly, the gaps become larger from Bleu-1 to Bleu-4 (from 1-gram to 4-gram), manifesting the superiority of the structured semantic representation in VSSI-cap. In summary, although VSSI-cap is designed for diverse captioning, the various but accurate visual semantic is well captured in the lexical and syntactic parsing results, which promotes the accuracy of VSSI-cap in the task of general image captioning.

**Evaluation on Diversity.** We compare the proposed VSSI-cap to the baseline and state-of-the-art methods on the diversity metrics in Tab. 3 shows. Despite there is a gap on the diversity when compared to the human captions, VSSI-cap achieves the best performance compared to other learning methods under most metrics, *e.g.*, the best 62.4% on mBleu (lower is better), which reflects the effectiveness of considering both the lexical and syntactic diversities in diverse image captioning, as well as the superiority of the proposed VarMI-tree based inferer on modeling these diversities.

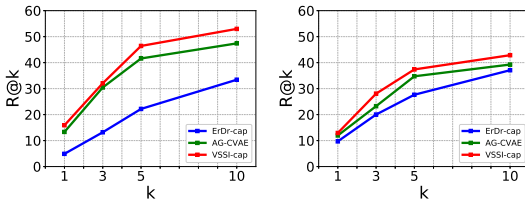

Figure 4: The recalls of image rankings for different methods. Given the generated caption queries, R@k is the ratio of correct images being ranked within the top k results. The left is based on the similarity (Left) between the generated caption $\tilde{S}$ and each image $I$, while the right is based on the log-likelihoods (Right) $P(\tilde{S}|I)$, computed in different methods.

Specially, VSSI-cap-L and VSSI-cap-S also achieve competitive performance. This manifests the significant roles of the lexical and syntactic diversities respectively. We conduct additional comparisons on the results with 10 generated captions (5 is default), where VSSI-cap (n=10) also outperforms AG-CVAE (n=10). We further retrieve the images of the generated captions in 5,000 images (randomly selected) by taking the captions as queries. The recalls of the ranking results are shown in Fig. 4, where our VSSI-cap is shown to provide more discriminative descriptions, outperforming others by a large margin across all cases. To qualitatively compare the performances on diversity, we output the results of VSSI-cap and the baselines of ErDr-cap and AG-CVAE (also a state-of-the-art) in Fig. 5. Clearly, VSSI-cap generates more diverse captions, which further demonstrates the superiority of the proposed VSSI-cap.

**Model Analysis.** It's a challenge to analyze the internal mechanism of the VAE-based structured encoder-inferer-decoder due to different vector spaces among 1) different node features, 2) different Gaussian functional parameters, and 3) different lexical/syntactic variables over different nodes. Fortunately, the parsing results of VP-tree can be assigned with different probability distributions (inputs of VarMI-tree) in each node to indirectly verify the effectiveness of the VarMI-tree, as shown in Fig. 6. Highly diverse captions are generated derived from different visual parsing trees with different lexical/syntactic probability distributions. This demonstrates the effectiveness of VarMI-

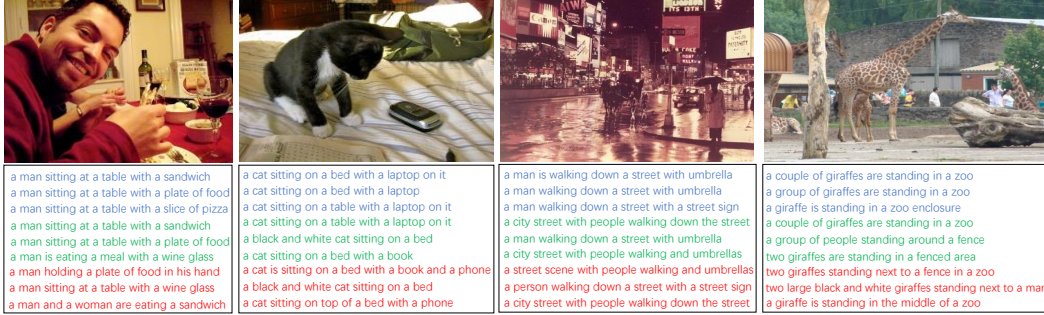

Figure 5: Visualization of diverse captions (top 3) generated by ErDr-cap (blue), AG-CVAE (green), and our VSSI-cap (red). More results are presented in the supplementary material.

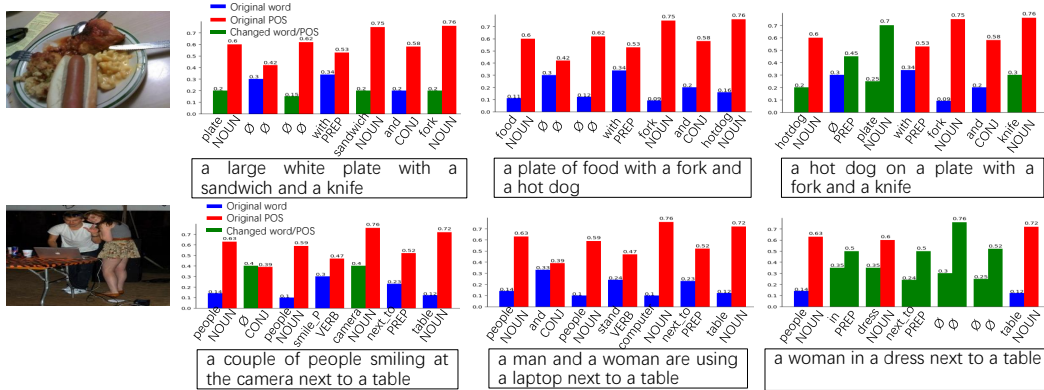

Figure 6: Internal view on the effectiveness of the VarMI-tree by changing its inputs explicitly, *i.e.*, assigning different lexical/syntactic probability distributions of each node from VP-tree (refer to Fig. 3 for the node index). The histograms of each example reflect different visual parsing trees with different probability distributions assigned in each node, where the middle is for the original parsing results (best-in-top3 is shown in each node) from VP-tree, while the left/right is for the parsing results partly changed from the middle mainly on word/POS. Captions are generated according to different visual parsing trees at the bottom.

tree on modeling the lexical/syntactic diversity and embedding them into caption generation in the proposed structured encoder-inferer-decoder.

## 5 Conclusion

In this paper, we exploit the key factors of diverse image captioning, *i.e.*, the lexical and syntactic diversities. To model these two diversities into image captioning, we propose a variational structured semantic inferring model (VSSI-cap) with a novel variational multi-modal inferring tree (VarMI-tree) on a structured encoder-inferer-decoder schema. Specially, conditioned on the visual-textual features from encoder, VarMI-tree models the lexicon and the syntax, as well as inferring their latent variables in approximate posterior inference guided by the visual prior. Reconstruction loss and KL-divergence are jointly estimated to optimize the VSSI-cap model to generate diverse captions. Experiments on benchmark dataset demonstrate that the proposed VSSI-cap achieves significant improvements over the state-of-the-arts.

## Acknowledgments

This work is supported by the National Key R&D Program (No.2017YFC0113000, and No.2016YFB1001503), Nature Science Foundation of China (No.U1705262, No.61772443, No.61572410, and No.61702136), Post Doctoral Innovative Talent Support Program under Grant BX201600094, China Post-Doctoral Science Foundation under Grant 2017M612134, Scientific Research Project of National Language Committee of China (Grant No. YB135-49), and Nature Science Foundation of Fujian Province, China (No. 2017J01125 and No. 2018J01106).

## Footnotes

[2]Compared to other generative models, VAE can represent richer latent variables, which can also be trained more easily.

[3]In some complex tasks, *e.g.*, image captioning, $q(z|x)$ is commonly termed as *inferer* to differ from the visual *encoder* CNN.

[4]Textual parsing and pruning preprocesses are conducted by following [13] to obtain the tree structure.

[5]http://cocodataset.org/#download

[6]https://github.com/karpathy/neuraltalk

[7]https://github.com/tylin/coco-caption

[8] https://github.com/cfh3c/NeurIPS19_VPtree_Dics

[9] https://github.com/yiyang92/vae_captioning

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
