[Supplementary Material]

# Variational Structured Semantic Inference for Diverse Image Captioning
## (Supplementary Material)

## A. Generated Diverse Captions

Additional examples of generated diverse captions are shown in Fig. 1, where, apparently, the captions in red are more diverse than the others. This is can be reflected on the syntactic and lexical diversities. For example, in the first image, the syntactic structures of the captions contain "...*VERB*...*PREP*...", "...*PREP*...", and "... *PREP*...*CONJ*...", which are more various than the combinations of {*VERB*,*PREP*} (in blue and green). The lexical diversity can be found in terms of various qualifiers and more entities, *e.g.*, *water*. However, the unspecific saliency and data bias may disrupt the models as shown in the last example, where only *street light/sign* is manly focused in the scene with the unspecific saliency. It would be an open problem for many works in future for further studies, including 1) balancing the lexical and syntactic diversities, 2) exploring the relation between the lexical/syntactic diversity and the data bias, and 3) considering the effect of the visual saliency on the lexical/syntactic diversity.

Figure 1: Visualization of diverse captions on MSCOCO dataset. The captions (top 3) are generated by ErDr-cap (blue), AG-CVAE (green), and our VSSI-cap (red).

## B. Supplement of Training Details

The loss curves are presented in Fig. 2, where the reconstruction loss of VSSI-cap converges better than AG-CVAE [8] when their KL terms go to the same level. This indicates the better representation of VSSI-cap than AG-CVAE on the diverse semantics. We adopt KL annealing method [34] for both VSSI-cap and AG-CVAE to reduce the KL vanishing. The curves of KL term values are shown in the zoom-out (right-top) and zoom-in (right-bottom) sub-figures of Fig. 2, where the KL term value of VSSI-cap converges faster and fluctuates less than AG-CVAE. These reveal the more reasonable (converges faster) and more various (fluctuates less) representation of the caption diversity when the prior and posterior get closed.

Figure 2: The loss curves on the reconstruction (left) and KL term (right) of two VAE based diverse image captioning schemes, *i.e.*, baseline AG-CVAE (blue) and our VSSI-cap (red). The right-bottom curve is the zoom-in of the right-top.

## C. Algorithm Flow

---

**Algorithm 1:** Training of VSSI-cap

---

**Input**: The image-caption pair set $\mathcal{D}_p$

1   Initialize the parameter sets $\theta$, $\phi^{(\ell)}$, $\phi^{(s)}$, and $\psi$;

2   **while** *not converged* **do**

3      **for** *I and S in* $\mathcal{D}_p$ **do**

4          Extract $\mathbf{v}$ and $\mathbf{e}$ from $I$ and $S$ via CNN and Parsing&Embedding, respectively;

5          Compute $\{\mathbf{c}^{(\ell)j}\}_{j=1}^M$ and $\{\mathbf{c}^{(s)j}\}_{j=1}^M$ as Eq. 4 in the pretrained VP-tree;
         `// For Posterior`

6          Compute $\{\boldsymbol{\mu}^{(\ell)j}\}_{j=1}^M$, $\{\boldsymbol{\mu}^{(s)j}\}_{j=1}^M$, $\{\boldsymbol{\sigma}^{(\ell)j}\}_{j=1}^M$, and $\{\boldsymbol{\sigma}^{(s)j}\}_{j=1}^M$ as Eq. 6 and Eq. 7;
         `// For Prior`

7          Compute $\{\boldsymbol{\mu}'^{(\ell)j}\}_{j=1}^M$, $\{\boldsymbol{\mu}'^{(s)j}\}_{j=1}^M$, $\{\boldsymbol{\sigma}'^{(\ell)j}\}_{j=1}^M$, and $\{\boldsymbol{\sigma}'^{(s)j}\}_{j=1}^M$ as Eq. 9 and Eq. 7; `// It's` `similar to the posterior.`
         `// For KL-divergence`

8          Approximate $D_{\mathrm{KL}}\big(q_{\phi^{(*)},\psi}(\mathbf{z}^{(*)j}|S,\mathbf{v},\mathbf{c}^{(*)j})\|p(\mathbf{z}^{(*)j}|\mathbf{c}^{(*)j})\big)$: `// ` $*$ ` denotes ` $\ell$ ` or ` $s$ `.`

9          $\log\left(\frac{\boldsymbol{\sigma}'^{(*)j}}{\boldsymbol{\sigma}^{(*)j}}\right) + \frac{\boldsymbol{\sigma}^{(*)j2}+\|\boldsymbol{\mu}^{(*)j}-\sum_{k=1}^{K^{(*)}}c_k^{(*)j}\boldsymbol{\mu}_k'^{(*)j}\|^2}{2\boldsymbol{\sigma}'^{(*)j2}}$;
         `// For Reconstruction`

10         Approximate $\mathbb{E}_d$ as Eq. 8 and Eq. 11;

11         Optimize Eq. 10 by stochastic gradient ascent method.

12      **end**

13 **end**

---

## D. Supplement of Model Analysis

We disentangle the syntactic and lexical effects in Tab. 1. After modifying lexicon/syntax respectively, we find: (1) the word and POS sequences are both changed (ED $\neq$ 0 and B-1,3 $\neq$ 100%), which reveals the inherent correlation between lexicon and syntax, (2) the change on word/POS sequences is bigger than that of POS/word, *i.e.*, higher ED and lower B-1,3, which indicates that the lexical/syntactic variables have more effect on the lexicon/syntax, and (3) the change on POS sequences is smaller than that on word sequences, which is probably due to smaller POS vocabulary.

Table 1: Consistency between the word/POS output sequences of the modified and original versions. Lexicon/syntax is modified by assigning random probabilities to words/POSs in VP-tree. Edit Distance (ED), Bleu-1 (B-1, %) and Bleu-3 (B-3, %) are taken to measure the consistency (B-1,3 are for the word-level evaluation).

| | Evaluation on word | | | Evaluation on POS | | |
|---|---|---|---|---|---|---|
| | ED | B-1 | B-3 | ED | B-1 | B-3 |
| Modify L. | 4.08 | 51.24 | 32.20 | 5.48 | 85.36 | 73.27 |
| Modify S. | 3.82 | 55.14 | 36.44 | 5.83 | 84.22 | 71.38 |