[Reviews · NeurIPS 2019]

Reviewer 1



Originality: The proposed approach using syntactic and lexical diversity modelling within the latent space to generate diverse image captions is novel. Quality: To establish that the generated captions are diverse, various standard diversity metrics are measured for the proposed method in Tab. 2. Some qualitative results demonstrating diverse captions and diversity conditioned on different visual parse tree probabilities is shown in Fig. 5 and 6. These experiments help justify the core components of the proposed approach. Clarity: The paper is well written and easy to follow. Careful illustrations in Fig. 2 and 3 are used as an aid while describing the proposed method. Significance: The proposed method has comparable accuracy to GAN/VAE-based diverse captioning counterparts, and it demonstrates better diversity scores. It is a valuable benchmark for future development in diverse captioning. Post-rebuttal comments -- The authors add additional experiments to measure edit distance between POS/words when the lexical or syntactic variables are changed, this is a good value add to earlier experiments. They address my concerns on experimental setup for baseline methods and should clarify this in the final submission. I remain positive about the contributions of the paper and maintain my earlier rating.

Reviewer 2



Originality: The paper is moderately original — the idea of splitting diversity into lexical and syntactic is interesting. However, the approach taken is derived from [13] and standard VAEs. Quality: The paper is technically correct and cites relevant work adequately. Clarity: While the writing is generally clear, the notation is a bit cluttered and hard to parse with the “.” and “..” superscripts for variables. Significance: The idea of splitting diversity into these two factors is both novel and interesting. The paper takes a reasonable route to model both these aspects and also shows that it leads to improvements (over at least other VAE-based methods, [3] and other RL methods perform better, see COCO leaderboard). Therefore, it is of reasonable interest to the community.

Reviewer 3



1. It is not clear to me that since each tree node has a local latent variable/representation, how does the model generalize to generate longer captions or paragraphs. 2. If possible, the authors should reorganize Section 3, as the current version is not easy to follow. 3. In general, the description framework seems a bit engineering. It would be helpful if the author could provide more abstract analysis and empirical discussion about the proposed framework.

[Author Response · NeurIPS 2019]

We thank all reviewers for their positive comments on idea novelty, technical quality, paper writing, and promising
directions. We respond to the concerns point-by-point as below.

**R#1.Q1. Apart from Fig. 6, would be useful to see if syntactic/lexical (S./L.) effects are disentangled.**

Following this suggestion, we provide quantitative comparisons to analyze the disentanglement of the L. and S. effects
in Tab. I. After modifying L./S., we have found: (1) the word and POS sequences are both changed (ED $\neq$ 0 and B-1,3
$\neq$ 100%), which reveals the inherent correlation between L. and S., (2) the change on word/POS sequences is bigger
than that of POS/word, *i.e.*, higher ED and lower B-1,3, which indicates that the L./S. variables have more effect on the
L./S., and (3) the change on POS sequences is smaller than that on word sequences, which is probably due to smaller
S. (POS) vocabulary. We will add the above results and discussions in our paper to further enrich the insights.

**R#1.Q2. On technical details.**

For the question whether beam search is used in [28][3], com-
mon image captioning methods [28] and [3] use beam search
for sampling. For the question which captions are used for
computing metrics in Tab. 1, we use likelihood to sample top
5 captions of each testing image, compute their metrics, and
choose the top caption for the evaluation in Tab. 1 to ensure a
fair comparison. We will clarify the above details in our paper.

Table I: Consistency between the word/POS output sequences of the modified and original versions. Lexicon/syntax (L./S.) is modified by assigning random probabilities to words/POSs in VP-tree. Edit Distance (ED), Bleu-1 (B-1, %) and Bleu-3 (B-3, %) [29] are taken to measure the consistency (B-1,3 are for the word-level evaluation).

|  | Evaluation on word | | | Evaluation on POS | | |
|---|---|---|---|---|---|---|
|  | ED | B-1 | B-3 | ED | B-1 | B-3 |
| Modify L. | 4.08 | 51.24 | 32.20 | 5.48 | 85.36 | 73.27 |
| Modify S. | 3.82 | 55.14 | 36.44 | 5.83 | 84.22 | 71.38 |

**R#2.Q1. On the originality compared to [13] & VAE.**

Thanks, we agree that our work does share certain intersection with [13] and VAEs. However, we rebut that our novelty
are fundamentally sufficient comparing to [13] and VAEs, which are detailed in two aspects:

*New problem formulation*: Conventional encoder-decoder depends solely on sampling to import randomness [28],
which limits the diversity among the outputs (see Tab. 2). VAE based encoder-decoder introduces the latent vari-
able and makes a two-stage inference for latent variables and words, which enhances the diversity (also see Tab. 2).
However, the latent variables in VAEs have a very general prior (standard Gaussians), which does not consider any
domain-specific knowledge. We argue that it may waste model capacity, and one should consider the unique problem
structures of image captioning instead of using the VAE as is. Therefore, we introduce the domain knowledge from
NLP and decompose the latent variables into lexicon and syntax variables. This fundamental change in the problem
representation is the core novelty of our approach. Under this guidance, it's a straightforward thinking to model a
structured variational inferrer with the assistance of VP-Tree [13] and the adaptation of VAE. However, we kindly ar-
gue that such assistance/adaptation is not our core contribution. One can replace VP-Tree with other visual structured
representations, or use generative models other than VAE. However, in order to directly demonstrate the effectiveness
of the core idea, we intentionally chose these straightforward adaptations, which help the readers directly catch our
main innovation and not get distracted by the complicated adaptation.

*New technical design*: VP-Tree [13] can provide the lexicon/syntax probabilities, which, however, does not involve
the construction and the prior/posterior inference of the latent variables (Sec. 3.2). For VAEs, though commonly used,
they never consider modeling the structured latent variables with the domain-specific knowledge, as well as jointly
optimizing the reconstruction and the prior-posterior distance with the structured latent variables (Sec. 3.3).

**R#2.Q2. Notations involving " ˙ " and " ¨ " can be replaced with sub/super-scripts $\ell$ and $s$.**

Thank you for this suggestion. We will modify accordingly in our paper.

**R#3.Q1. Confusion on model generalization for longer captions or paragraphs.**

Thanks for this inspiring question. VarMI-tree can be easily expanded to the case with more tree nodes for long
captions. Our model itself has no such limitation on caption length, and we just set the node number as 7 to cover
the captions in the COCO dataset. As for image paragraph description, it is quite different from the sentence-level
captioning due to different topics of sentences [A][B]. However, as long as the topic feature of each sentence can
be extracted from RNN [A][B] to construct VarMI-tree, it does not restrict our model to be generalized to image
paragraph description.

[A] M. Chatterjee *et al.* Diverse and Coherent Paragraph Generation from Images. ECCV 2018.
[B] J. Krause *et al.* A Hierarchical Approach for Generating Descriptive Image Paragraphs. CVPR 2017.

**R#3.Q2. Reorganizing Section 3 for easier follow.**

Thanks for the suggestion. We will strengthen the method overview with a table of notations and an algorithm flow.

**R#3.Q3. Providing more abstract analysis and empirical discussion.**

Thanks for this suggestion. Please kindly refer to our response to Q1 of R#2, which will be added in our paper.

[Meta-Review · NeurIPS 2019]

After considering the author response and discussing the submission, the reviewers all voted to accept the submission. The approach presented puts forward a novel framing for caption diversity and the empirical evaluation supports the paper's contributions. The document as a whole could use additional clarity so I urge authors to spend time revising it to broaden the impact of this work.